Subject Areas:
computer modelling and simulation/ mathematical modelling/mathematical physics

Keywords:
migration, refugees, social simulations, computational social science, science of cities

Author for correspondence:
Anand Sahasranaman
e-mail: a.sahasranaman15@imperial.ac.uk

# Rapid migrations and dynamics of citizen response

Anand Sahasranaman[1,2] and Henrik Jeldtoft Jensen[1]

[1]Department of Mathematics and Centre for Complexity Science, Imperial College London, London SW7 2AZ, UK
[2]Division of Mathematics and Computer Science, Krea University, Sricity, Andhra Pradesh 517646, India

 AS, 0000-0002-9613-8041; HJJ, 0000-0002-5398-3288

One of the pressing social concerns of our time is the need for meaningful responses to migrants and refugees fleeing conflict and environmental catastrophe. We develop a computational model to model the influx of migrants into a city, varying the rates of entry, and find a nonlinear inverse relationship between the fraction of resident population whose tolerance levels are breached due to migrant entry and the average time to such tolerance breach. Essentially, beyond a certain rate of migrant entry, there is a rapid rise in the fraction of residents whose tolerances are breached, even as the average time to breach decreases. We also model an analytical approximation of the computational model and find qualitative correspondence in the observed phenomenology, with caveats. The sharp increase in the fraction of residents with tolerance breach could potentially underpin the intensity of resident responses to bursts of migrant entry into their cities. Given this nonlinear relationship, it is perhaps essential that responses to refugee situations are multi-country or global efforts so that sharp spikes of refugee migrations are equitably distributed among nations, potentially enabling all participating countries to avoid impacting resident tolerances beyond limits that are socially sustainable.

## 1. Introduction

We live in a time of significant global conflict. The UNHCR reports that millions of people have been displaced by conflict just in the recent past—with 6.6 million internally displaced Syrians, 2.5 million refugees from South Sudan, 2.1 million Iraqi refugees, 723 000 Rohingyas fleeing Myanmar and 200 000 displaced persons in Yemen only partially illustrating the extent of strife around the world [1]. In addition to displacements due to conflicts, we are likely to also see increasing forced displacement on account of climate change, with estimates that there could be as many as 200 million people displaced worldwide on account of global warming by 2050 [2].

One of the consequences of these displacements is the increasing flow of refugees into neighbouring countries or the

undertaking of even longer, more dangerous journeys into cities of the developed world. For instance, most of the over 700 000 Rohingya refugees have sought refuge in Bangladesh [3], 3.5 million Syrian refugees have relocated to Turkey [4], close to 1 million Syrian refugees live in Lebanon [5], and Europe continues to remain a destination for those displaced in the Middle East and Africa [6]. Additionally, refugees have been found to predominantly move into cities in their countries of destination [4,7]. As these large migration flows have occurred over compressed time frames, there is also growing evidence of backlash at refugees by residents in these nations. About 400 000 refugees entered Turkey in 2013, close to 1 million entered in 2014 and 2015, 600 000 came through in 2017 [4], and despite attempts by the Turkish government at resettling refugees, there appears to be growing resentment towards Syrian refugees in Turkey [8]. Similarly, 700 000 refugees entered Lebanon in 2013–2014 and over 200 000 in 2014–2015 [5], bringing the refugee population in Lebanon to over 20% of the total population, and resulting in a rise in anti-refugee sentiment that has now manifested in refugee evictions from Lebanese cities [9]. Over 1.6 million refugees have arrived in Europe over the past 4 years (2015–2018), with significant annual variations—out of this total, over 1 million (or 62.3%) arrived in 2015 alone, while 184 000 (or 11.3%) arrived in 2017 [10]. Refugee arrivals show significant temporal variance and average arrivals can be an extremely misleading indicator of arrival behaviour. The sharp uptick in refugee arrivals in Europe in 2015 was manifested in the fact that asylum applications in eight European countries were more than 2000 per million inhabitants [11]. The situation was especially stark in Germany, which received over 475 000 asylum applications in 2015 (or approx. 5814 per million inhabitants), but even more dramatically, saw the arrival of more than 1 million refugees into its cities that year [11]. Montero & Baltruks [11] pinpoint perhaps the most pertinent concern with such variability in arrivals, which is the effect that sudden and potentially significant increases in urban population could have on society and public services in cities.

There has been some work on how the entry of migrants has impacted the attitudes of extant city residents to the new arrivals. Wike *et al*. [12] find that post the recent surge of Syrian refugees in Europe, over half the population in eight out of 10 European countries surveyed believe that incoming refugees increase the likelihood of terrorism in their country, and over half the population in six of the countries see refugees as a burden on jobs and social benefits. The rise of these attitudes, they argue, can be linked to the growing anti-immigrant rhetoric among political parties in Europe. Milanović [13] further substantiates this phenomenon by providing evidence of the significant and rapid increase in vote share of populist, nativist political parties across Europe between 2000 and 2017—for instance, vote shares of parties like AfD in Germany, Jobbik in Hungary and Danish People's Party in Denmark rose from 0%, 2% and 7% in 2000 to 13%, 20% and 21% by 2017, respectively. It is therefore plausible that the rapidity of migration could indeed impose a significant influence on the intensity of resident attitudes to migrants.

It is this specific concern that drives our work here—how does the change in the rate of migration into cities affect the tolerance of resident citizens to incoming migrants? This, in the present context, is a critical question confronting cities as they devise appropriate strategies for dealing with increasing influx of migrants, while still maintaining social and economic stability.

The Schelling model [14] offers a suitable framework to explore this question. The Schelling model was proposed by Thomas Schelling in 1969 to explain the emergence and persistence of racial segregation in American cities. It was possibly the first agent-based model in the social sciences and the dynamics of the model were based on the tolerance levels ($\tau$) of individual agents of two races. The agent tolerance level was defined as the maximum fraction of neighbours of the opposite race a given agent was willing to tolerate in its neighbourhood before attempting to move out. At each iteration, an agent assessed its satisfaction with the racial composition of its neighbourhood and if unsatisfied, attempted to move to a neighbourhood where the condition was satisfied. Very quickly, the dynamics were found to settle into a segregated equilibrium. The central insight of the model is that even small preferences for like neighbours at an individual level (meaning individual preferences for integration rather than segregation) yield the emergence of segregated patterns at a collective level. The Schelling model is found robust to parameter specifications such as neighbourhood definition, agent preferences and agent choice functions [15–19]. While there have been diverse explorations of the Schelling model [20–23], modelling migration into a Schelling framework has remained a lesser explored area. Urselmans [24] modelled the entry of a migrant population into a city of residents and found that overall agent happiness (in terms of fraction of residents/migrants in the neighbourhood) converges over time, irrespective of the size and rate of migration. On the basis of the Schelling model, we had explored the phenomenon of migration into a city and its effect on 'dual' segregations—ethnic segregation and wealth segregation—and found the possibility of a trade-off

between these segregation tendencies as agents were progressively allowed to move into neighbourhoods they could not afford [25].

Our objective here is to model the entry of migrants and study its impact on the tolerance levels of residents in the city. Specifically, we are interested in varying the rates of migrant entry to simulate scenarios of sharp, significant bursts of migration. While we would expect long-term outcomes on segregation across scenarios to converge, our particular interest is in the transient dynamics of emergent neighbourhood configurations as a response to the rapidity and intensity of migrant entry into the city.

## 2. Model definition and specifications

Our model is largely drawn from previous work we had done on the impact of migration on the emergence of dual segregations, namely ethnic segregation and wealth segregation [25]. As in that model, we consider a 'city' with $M$ neighbourhoods, with each neighbourhood $i$ ($i \in 1, \ldots, M$) initially populated by $p$ agents. This initial set of agents is defined as the resident population (or residents) of the city. Each resident agent has a single attribute, namely resident tolerance level ($\tau_{\mathrm{res}}$). As with the tolerance level in the original Schelling model, we define the resident tolerance level ($\tau_{\mathrm{res}}$) as the maximum fraction of migrant population in a neighbourhood that a resident agent is willing to tolerate. Sethi & Somanathan [26] state that studies on neighbourhood composition preferences reveal that people have a preference for some degree of integration with a bias for like neighbours; they also refer to an American survey work which reveals that white Americans, on average, have lower tolerance levels than African Americans. We set $\tau_{\mathrm{res}} = 0.25$ for all resident agents in the simulation.

At each iteration (or time step, $t$) of the model, Mig($t$) migrants attempt to enter the city. This is determined as a fraction $g$ of the extant population of the city at the end of time step ($t - 1$), $P(t - 1)$ (equation (2.1))

$$\mathrm{Mig}(t) = gP(t - 1). \tag{2.1}$$

We vary $g$ to change the quantum of migrants attempting entry into the city, and for the simulations use the following values for $g$: $g = 0.005, 0.007, 0.01, 0.015, 0.02, 0.03, 0.04, 0.05$. We simulate over this wide range of values so as to mimic both orderly, predictable growth over time and bursty growth over short time spans that could, for instance, be caused by crises such as climate change and conflict. We also restrict total population growth to 1.5 times the initial population (beyond which level, at each iteration, zero migrants attempt to enter the city), as our objective is to study the transient dynamics resulting from variation in speed of migration into the city.

Like resident agents, migrant agents are also defined by a single characteristic, migrant tolerance level ($\tau_{\mathrm{mig}}$), which is the maximum fraction of resident agents a migrant agent is willing to tolerate in a neighbourhood. For the simulations, we set $\tau_{\mathrm{mig}} = 0.75$.

The movement of migrants into the city is predicated on the population of residents in the neighbourhood they attempt to move into. We base this design choice on the theory of spatial assimilation [27], according to which, immigrants first settle in homogeneous ethnic enclaves in cities, but as their socioeconomic condition improves, they seek to move into wealthier neighbourhoods of the dominant resident majority. Migrant entry into co-ethnic enclaves has also been revealed by detailed sociological work on the distinct neighbourhoods of immigrants in cities [24], as well as ethnic concentration in cities with migration being a primary contributor [28]. Spatial relatedness between incoming immigrants and existing communities has also been found to be a useful measure in analysing the dynamics of regional population change [29]. Specifically, therefore, we model migrant entry into the city as being stochastic, based on the ratio of population of residents ($P_i^{\mathrm{res}}$) to total population ($P_i^{\mathrm{tot}}$) in a randomly chosen cell $i$ that the migrant agent attempts to enter (equation (2.2)). This choice echoes our previous work on concurrent ethnic and wealth segregation in cities [25], and yields increasing probability of migrant entry with decreasing fractions of resident agents in neighbourhoods

$$p_{\mathrm{entry}}(i) = \exp\left(-\beta_{\mathrm{in}} \frac{P_i^{\mathrm{res}}}{P_i^{\mathrm{tot}}}\right), \tag{2.2}$$

where $\beta_{\mathrm{in}}$ is a calibrating factor that determines $p_{\mathrm{entry}}$. In the limit $\beta_{\mathrm{in}} \to \infty$, no migrants enter the city, while in the limit $\beta_{\mathrm{in}} \to 0$, all potential Mig($t$) migrants enter the city. For the simulations, we set $\beta_{\mathrm{in}} = 1$.

Once migrant entry for the iteration (time step, $t$) is completed, we model agent movement within the city. At each time step $t$, we randomly choose $P(t)$ agents to attempt movement, thus ensuring that, on average, every agent has the opportunity to move at each time step. Movement within the city is based on

**Table 1.** Computational model parameters.

| parameter | value |
|---|---|
| number of neighbourhoods ($M$) | 20 |
| number of resident agents | 1000 |
| resident tolerance level ($\tau_{res}$) | 0.25 |
| migrant tolerance level ($\tau_{mig}$) | 0.75 |
| migration rate ($g$) | 0.005, 0.007, 0.01, 0.015, 0.02, 0.03, 0.04, 0.05 |
| maximum population (% of initial population) | 150% |
| number of iterations | 200 |
| $\beta_{in}$ | 1 |

the classic Schelling model implementation, where each randomly chosen agent $a$ attempts to move out of its neighbourhood $i$ if its tolerance level is breached. If there is indeed a breach, a random neighbourhood $j$ is chosen and the agent moves to $j$ if its tolerance level condition is satisfied. If not, $a$ stays back in its current neighbourhood $i$.

The update process in the model is sequential, and all model variables are updated at the end of each agent action. We model a city with $M = 20$ neighbourhoods. The dynamics are run over 200 iterations, which means that each resident agent, on average, is sampled 200 times, and all migrant agents, on average, are sampled as many times as the number of time steps (iterations) they spend in the city. Finally, for each of the parameter values of $g$ (detailed in table 1), we run an ensemble of 30 runs of the model.

It is important to point out that as our focus is on examining the transient dynamics due to bursts of migration, we do not consider the possibility of migrants transitioning into residents over time. While this is indeed a plausible phenomenon over longer time horizons, given our interest here, migrants entering the city remain migrants for the course of the dynamics.

We seek to understand the impact of migration into the city on the tolerance of resident agents. We use three distinct measures for this purpose: (i) current fraction of resident breach ($B_c$); (ii) fraction of residents ever breached ($B_f$), and (iii) average time to first breach ($T_\tau$).

The current fraction of resident breach ($B_c$) is simply the ratio of number of resident agents whose tolerance levels ($\tau_{res}$) are breached in their current neighbourhoods ($R_{cb}$) to the total number of resident agents in the city ($R_{tot}$) at the end of each iteration $t$ (equation (2.3))

$$B_c(t) = \frac{R_{cb}(t)}{R_{tot}}.$$

(2.3)

The fraction of residents ever breached ($B_f$) is the ratio of number of resident agents whose tolerance levels ($\tau_{res}$) have ever been breached ($R_b$) to the total number of resident agents ($R_{tot}$) at each iteration $t$ (equation (2.4)). This is different from $B_c$, as it accounts for all resident agents whose tolerance levels have ever been breached at any point in the dynamics, even if their tolerance levels are not breached at the current time.

$$B_f(t) = \frac{R_b(t)}{R_{tot}}.$$

(2.4)

The average time to tolerance breach ($T_\tau$) is the number of iterations or time steps, on average, taken for a resident agent to have its tolerance level ($\tau_{res}$) breached for the first time. For each resident agent $r$, we compute the number of iterations taken for the agent to have its tolerance limit breached ($T_\tau(r)$) and average this across all $R_b$ resident agents whose tolerance levels have been breached at the end of the simulations (equation (2.5)).

$$T_\tau = \frac{1}{R_b} \sum_{r=1}^{R_b} T_\tau(r).$$

(2.5)

We also measure the extent of segregation between residents and migrants that emerges at the end of each iteration using the Cell Segregation Indicator (CSI), a measure proposed by Gargiulo et al. [30] which encapsulates the local properties of the system by averaging cell-level heterogeneity over all non-empty

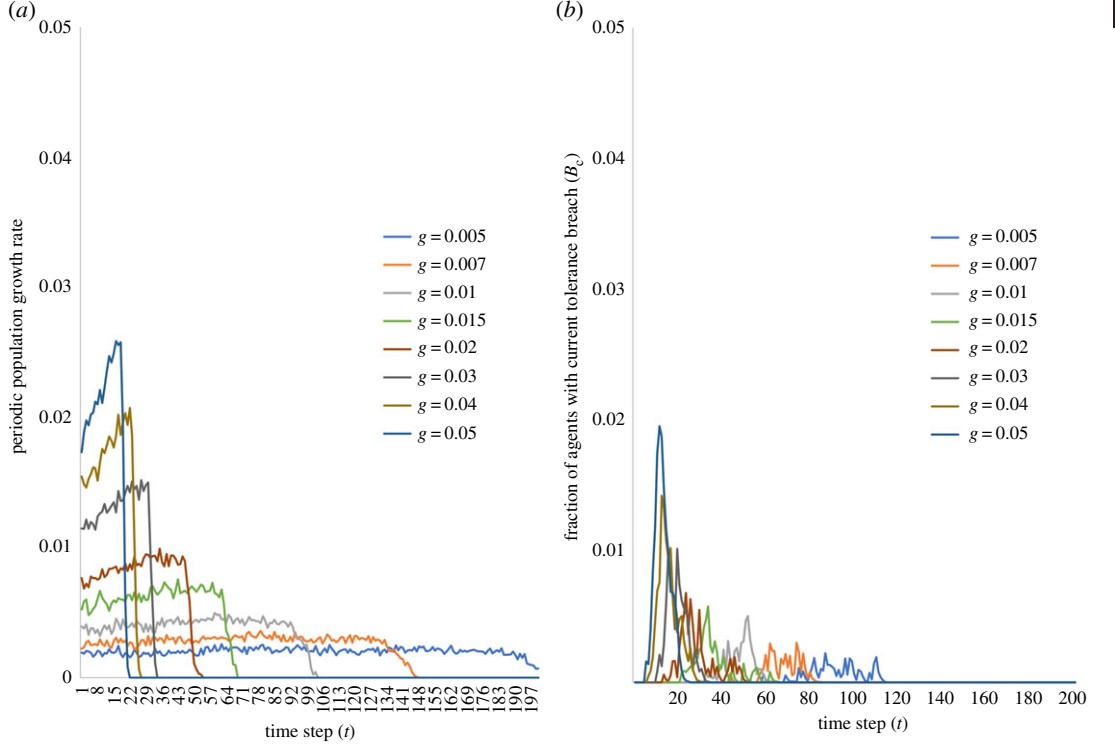

**Figure 1.** Population change and current tolerance breach. (*a*) Periodic growth rates of population over time for different *g*. As *g* increases, we see bursts of migration at increasingly higher rates and for progressively lesser time. (*b*) Current fraction of resident breach $B_c(t)$ over time *t*. $B_c$ remains low for even the highest values of *g*, with a peak of 1.96% attained during the dynamics for $g = 0.05$. Individual curves in both the figures represent different values of *g*: $g = 0.005, 0.007, 0.01, 0.015, 0.02, 0.03, 0.04, 0.05$.

cells. Lower CSI values indicate higher heterogeneity. If $f(r_t^i)$ and $f(m_t^i)$ are the fractions of resident and migrant agents (as proportions of total resident and total migration populations, respectively) in cell *i* at time step *t*, the cell segregation indicator CSI at time *t* is as follows:

$$\text{CSI} = \frac{1}{\sum_{i|f(r_t^i)+f(m_t^i)>0} 1} \sum_{i|f(r_t^i)+f(m_t^i)>0} \frac{|f(r_t^i) - f(m_t^i)|}{f(r_t^i) + f(m_t^i)}. \tag{2.6}$$

## 3. Dynamics of tolerance breach

Before we assess the outcome measures, we look at the population increase patterns for different values of *g*. As we would expect, there is significant variation in population paths (figure 1*a*). For instance, for $g = 0.005$, the periodic population growth rate is fairly stable over the 200 iterations at $\simeq 0.20\%$, while for $g = 0.01$, it is $\simeq 0.40\%$ for the first 103 iterations and 0% thereafter, and finally for $g = 0.05$, the periodic growth rate is $\simeq 2.02\%$ over the first 21 iterations and 0% thereafter, indicating a highly bursty growth pattern.

We first assess the outcomes on current fraction of resident breach, $B_c(t)$, across all values of *g* at the end of each iteration *t* of the model. We find that irrespective of the rate of migrant entry into the city, $B_c(t)$ does not exceed 2% (achieved for $g = 0.05$) at any point during the dynamics (figure 1*b*). This appears to suggest that even rapidly increasing the rate of agent entry into the city leaves over 97% of the resident population satisfied with their current neighbourhood composition at all times. Given this outcome, it could be argued that the rate of migrant entry is potentially not a significant determinant of resident attitudes to migrants, even when residents have low tolerance levels as in this case ($\tau_{res} = 0.25$).

However, this notion becomes difficult to sustain when we consider the behaviour of the fraction of residents ever breached, $B_f(t)$, over time *t*. Figure 2*a* shows us that the amount of time for the first tolerance breach to occur decreases progressively with increase in *g*—for instance, when $g = 0.005$, it takes 58 time steps for the first agent breach to occur, while for $g = 0.01$ it takes 32 time steps, and finally for $g = 0.05$, it takes a mere five time steps. Secondly, the value at which $B_f$ settles at the end of simulations for each value of *g* increases with increasing *g*—only 5.3% of resident agents have their

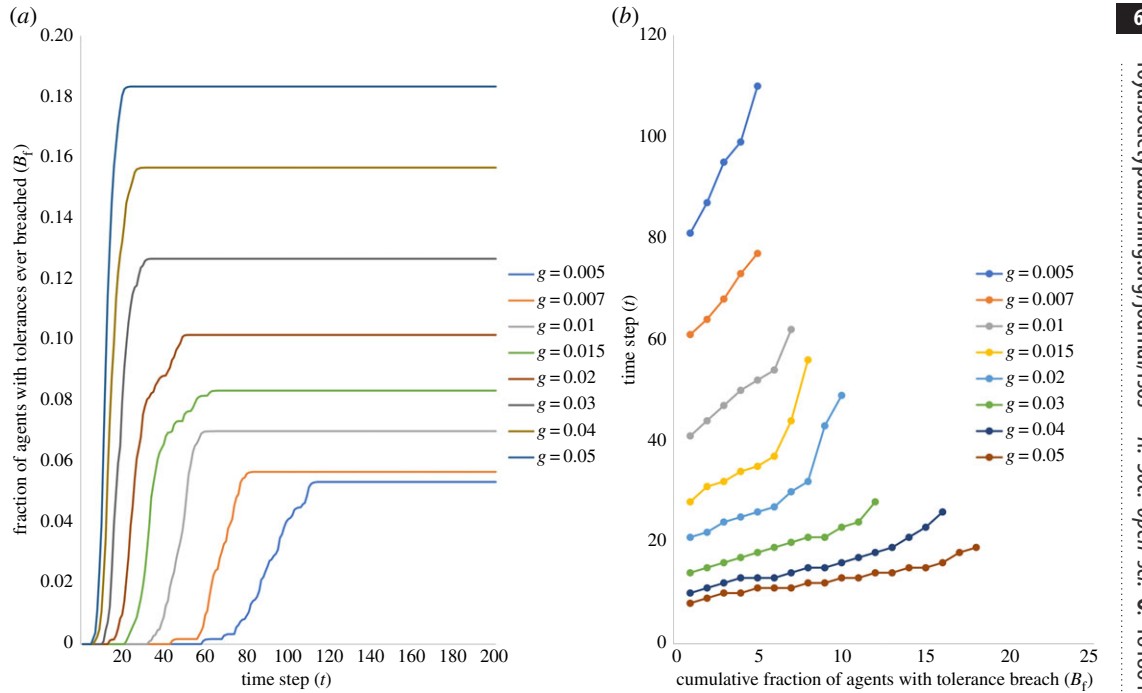

**Figure 2.** Tolerance breach. (*a*) Fraction of residents whose tolerance has ever been breached $B_f(t)$ over time $t$. The time taken for the first tolerance breach to occur decreases with increasing $g$, and the final $B_f(t)$ values (at $t = 200$) increase with $g$. (*b*) Time steps ($t$) required for every percentage point increment in $B_f$. The time taken for increasing fractions of agents' tolerances to be breached decreases with increasing $g$. Individual curves in both the figures represent different values of $g$: $g = 0.005$, 0.007, 0.01, 0.015, 0.02, 0.03, 0.04, 0.05.

tolerances breached in the course of the dynamics for $g = 0.005$, while the corresponding fraction is 18.3% for $g = 0.05$. Finally, the rising steepness of the curves also indicates that the time taken for increasing fractions of agents to have their tolerance levels breached decreases with $g$. This is more clearly illustrated in figure 2*b* where we find that for $g = 0.005$, it takes from 58 time steps (when the first breach occurs) to 81 time steps for 1% of resident agents' tolerance levels to be breached, and 110 time steps for 5% resident tolerance breach—meaning that it takes a total of 52 time steps from the first breach to 5% of agents having their tolerance levels breached. For $g = 0.05$, given the first breach occurs at 5 time steps, it takes a mere 14 more time steps for 18% of resident agents' tolerances to be breached. Therefore, not only are more resident agents' tolerances being breached, this is happening at a quicker rate.

When we plot the final $B_f$ values for different $g$ against the average time to tolerance breach for resident agents ($T_\tau$), we find a nonlinear inverse relationship between these measures (figure 3). Essentially, even as $g$ slowly increases initially, from 0.005 to 0.007, we find that there is a sharp drop in $T_\tau$ from 91.3 time steps to 67.5 times steps, even though $B_f$ only increases marginally from 5.33% to 5.67%. Beyond $g = 0.01$, the drop in $T_\tau$ becomes less steep, but the drop in $B_f$ becomes much steeper. For instance, between $g = 0.01$ and 0.02, $T_\tau$ drops from 47.7 to 28.4 time steps, but $B_f$ rises from 7% to 10.17%. While we indeed expect that $B_f$ would increase with $g$, and $T_\tau$ decrease with $g$, the nonlinear nature of this curve suggests something critically important. It suggests the possibility that beyond a certain value of $g$, the inverse relationship between $B_f$ and $T_\tau$ could yield socially unsustainable configurations—meaning that in such scenarios of short and sharp migration bursts (corresponding to $g > 0.01$), very high proportions of resident agents have their tolerances breached in very short time spans, potentially resulting in urgent and high intensity responses of resident agents to the entry of migrants.

We now seek to understand the dynamics underlying the nonlinear relationship between $B_f$ and $T_\tau$. As is evident from figure 2*b*, the time taken for the first tolerance breach to occur, as well as the speed of subsequent tolerance breaches decreases progressively with $g$. We explore this phenomenon by studying the change in $B_f$ as a function of the increase in overall system population due to migration. Figure 4 plots this relationship and suggests two distinct regions in this relationship. In the initial part of the curve, we find that the extent of increase in population required for $B_f$ to increase from 0% to 5% is similar for all $g$. For instance, when $B_f$ is 1%, we find that population increase across all scenarios of $g$ is $\simeq$18%, and when $B_f$ is 5% the population increase is $\simeq$24%. However, for $B_f > 6\%$, the nature of this relationship changes,

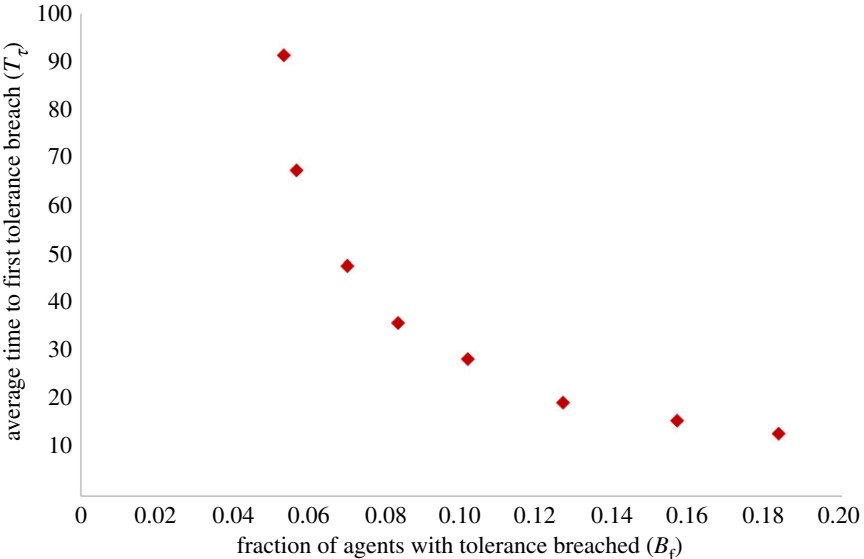

**Figure 3.** Average time to tolerance breach ($T_\tau$) versus total fraction of residents ever breached ($B_f$). There is an inverse, nonlinear relationship between $T_\tau$ and $B_f$.

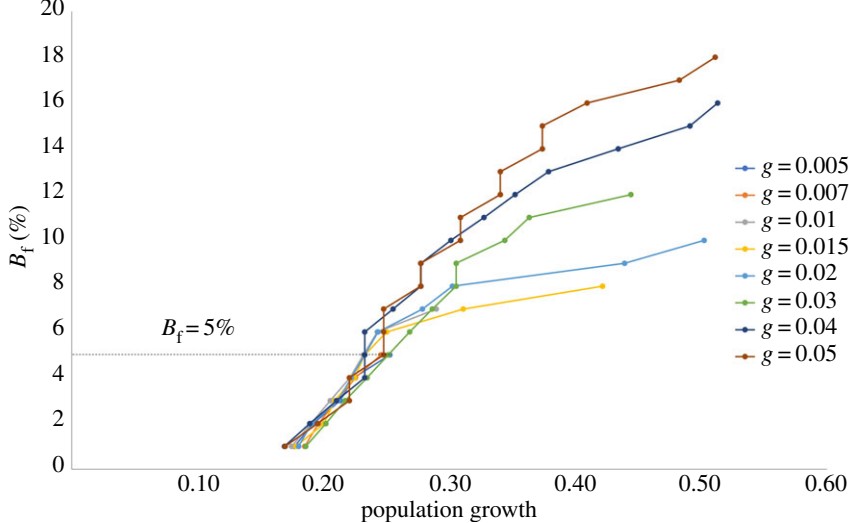

**Figure 4.** Total fraction of residents ever breached ($B_f$) versus population increase (as proportion of initial population). Dashed line: $B_f = 5\%$. Increase in population required for $B_f$ to increase from 1% to 5% is similar across all $g$, but as $B_f$ increases beyond this, lesser migrant influx is required to cause higher $B_f$ with increasing $g$.

and we find that it takes lesser increase in population for a greater $B_f$ to manifest, with increasing $g$. The population increase required to achieve $B_f = 9\%$, for example, is 44% for $g = 0.02$, 31% for $g = 0.03$, and 28% for $g = 0.05$. Similarly, at $B_f = 12\%$, population increase is 44% for $g = 0.03$, 35% for $g = 0.04$, and 34% for $g = 0.05$. And finally, at $B_f = 15\%$, population increase is 49% for $g = 0.04$, and 37% for $g = 0.05$.

What this behaviour appears to suggest is that as population increases after the first tolerance breaches occur (in the region $B_f = 1\%$–5%), residents are able to arrange themselves into satisfactory neighbourhoods rapidly enough such that even at higher bursts of migrant entry, it requires similar population increases across all $g$ for similar $B_f$ to manifest. It is important to note that the time taken for these equivalent population increases declines with $g$, as is apparent from figure 2b. The change in this relationship for $B_f > 5\%$, is an indication that resident agents find it increasingly difficult to rearrange themselves into satisfactory neighbourhoods quickly enough given the rates of increase in population at higher $g$, thus resulting in the need for progressively lower population increase (with increasing $g$) to yield similar $B_f$. We posit that it is indeed the twin-regimes evident in the relationship between population and $B_f$ that cause the nonlinearity in the relationship between $T_\tau$ and $B_f$.

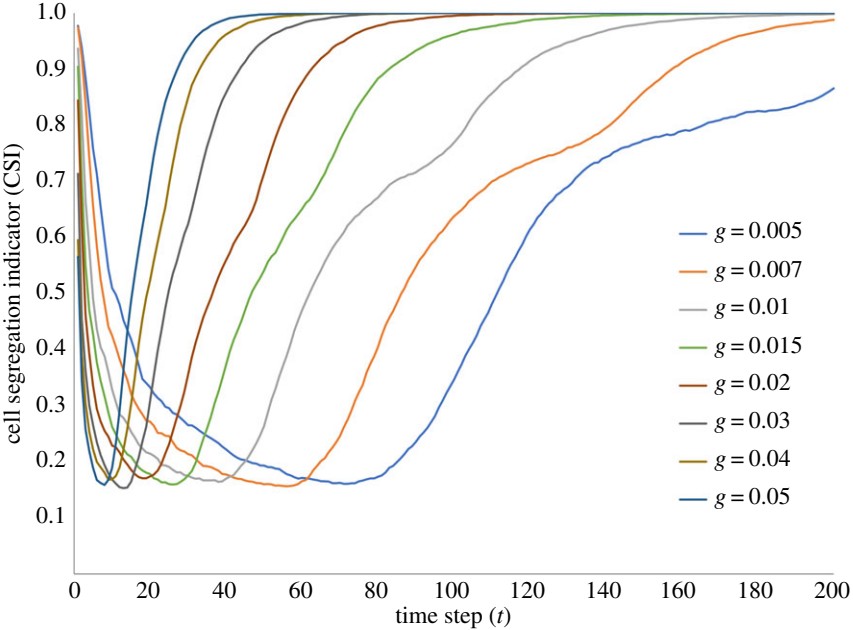

**Figure 5.** Cell segregation indicator (CSI) over time. When migrant influx begins, segregation declines and does so until resident tolerances begin being breached, at which point segregation starts rising monotonically.

# 4. Dynamics of segregation

Finally, if we plot outcomes on segregation between migrants and residents, we find that over the long term, segregation emerges and persists, just as predicted in the original Schelling model [14] and confirmed by Urselmans [24]. However, as figure 5 makes clear, while there is long-term convergence in segregation across all values of $g$, there are intermittent dynamics resulting in a lowering of segregation when migrant entry begins and for as long as the tolerance levels of resident agents remain unbreached. It is only when their tolerance levels are breached and they begin to spatially reconfigure themselves into more satisfactory neighbourhoods (i.e. with more like neighbours), that we see a monotonic rise and ultimate convergence in segregation over time for all values of $g$. This curve suggests the possibility that if migrant influx is managed below certain levels (different for different $g$), it is possible that the tolerances of resident agents remain largely unbreached (low $B_f$) and the level of segregation is significantly reduced as well.

Therefore, while our findings confirm the Schelling intuition that agents segregate themselves into neighbourhoods over time, it also appears to suggest that the transient dynamics caused by duration and size of migrant entry could yield a nonlinear inverse relationship between the fraction of agents whose tolerances are breached and the average time for such breach to occur, resulting in potentially unsustainable resident–migrant spatial configurations when bursts of migration beyond a certain size occur over relatively short durations.

We test the robustness of these results by varying model parameters and present the results in the electronic supplementary material. Specifically, we test sensitivity to: (i) maximum population limit; (ii) resident tolerance levels ($\tau_{res}$); (iii) sensitivity of agent choice function; and (iv) ease of migrant entry into city ($\beta_{in}$). We find the model outcomes to be robust to these perturbations.

# 5. Analytical model

We also attempt to create a formal analytical description of the phenomenology discussed so far, but the globally coupled nature of the dynamics make a master equation based formalism difficult in this case. However, we present a simplified analytical model to describe the dynamics. The simplifying assumptions make it impossible to replicate the entire range of phenomenology observed in the computational interpretation, but we believe it at least offers the possibility of qualitative comparison.

Consider a city of $M$ neighbourhoods ($C_1, C_2, \ldots, C_M$). The total population of residents in the city is $N$, equally distributed across all neighbourhoods. Each neighbourhood has a population of $N/M$. All residents have tolerance level, $\tau_r$, defined as follows:

$$\tau_r = \frac{j}{N/M}. \tag{5.1}$$

At each iteration, $Mx$ migrants enter the city, and are equally distributed across all neighbourhoods, resulting in $x$ migrants per neighbourhood per iteration. This corresponds with $g$ in the simulations, with the significant simplification that $x$ is a deterministic entry population per neighbourhood per iteration, while $g$ is a fraction of the city's population that attempts entry into the city at each time period, with actual entry being probabilistic (as per equation (2.2)).

Migrant agents are defined as completely tolerant, $\tau_m = 1$. Once migration occurs for an iteration, resident agents assess their neighbourhoods (based on $\tau_r$) and determine whether to stay or move. This process occurs sequentially by neighbourhood, starting with $C_1$ and ending with $C_M$ at each iteration. If $\tau_r$ is unsatisfied for a given neighbourhood $C_p$ ($1 < p < M$), then all resident agents in $C_p$ are equally distributed among the other neighbourhoods that follow it in sequence ($C_{p+1}$ to $C_M$).

Given that neighbourhoods are deterministically sequenced, the first neighbourhood where agent tolerances are breached is $C_1$. The number of migrants in $C_1$ for the breach to occur ($I_1$), the time to this breach ($T_1$), and the fraction of residents whose tolerance is breached ($B_1$) are given by (equation (5.2))

$$I_1 = j, \tag{5.2}$$

$$T_1 = \frac{I_1}{x} \tag{5.3}$$

$$\text{and} \quad B_1 = \frac{1}{M}. \tag{5.4}$$

All $N/M$ residents whose tolerances are breached from $C_1$ are now equally distributed among cells $C_2$ to $C_M$, each cell getting $N/M(M-1)$ additional residents. Given the occurrence of the first breach, the next breach occurs when additional migrants enter the city and breach the tolerance of residents in $C_2$. The number of additional migrants required to cause this breach ($I_2$) is obtained as solution to the following equation:

$$\frac{j}{N/M} = \frac{I_1 + I_2}{N/(M-1)}. \tag{5.5}$$

Therefore, the tolerance breach of a further $N/M$ residents (who were the initial residents of $C_2$) is caused by the entry of $I_2$ migrants, in time $T_2$, given by (equation (5.6))

$$I_2 = \frac{j}{M-1}, \tag{5.6}$$

$$T_2 = \frac{I_1 + I_2}{x} \tag{5.7}$$

$$B_2 = \frac{2}{M}, \tag{5.8}$$

where, $B_2$ is the total fraction of residents whose tolerances have been breached thus far.

In general, the statistics describing the tolerance breach of the $u$th set of $N/M$ residents, in cell $C_u$, are given by (equation (5.9))

$$I_u = \frac{Mj}{(M-u+1)(M-u+2)}, \tag{5.9}$$

$$T_u = \sum_{q=1}^{u} \frac{I_q}{x} \tag{5.10}$$

$$\text{and} \quad B_u = \frac{u}{M}. \tag{5.11}$$

royalsocietypublishing.org/journal/rsos　　R. Soc. open sci. **6**: 181864

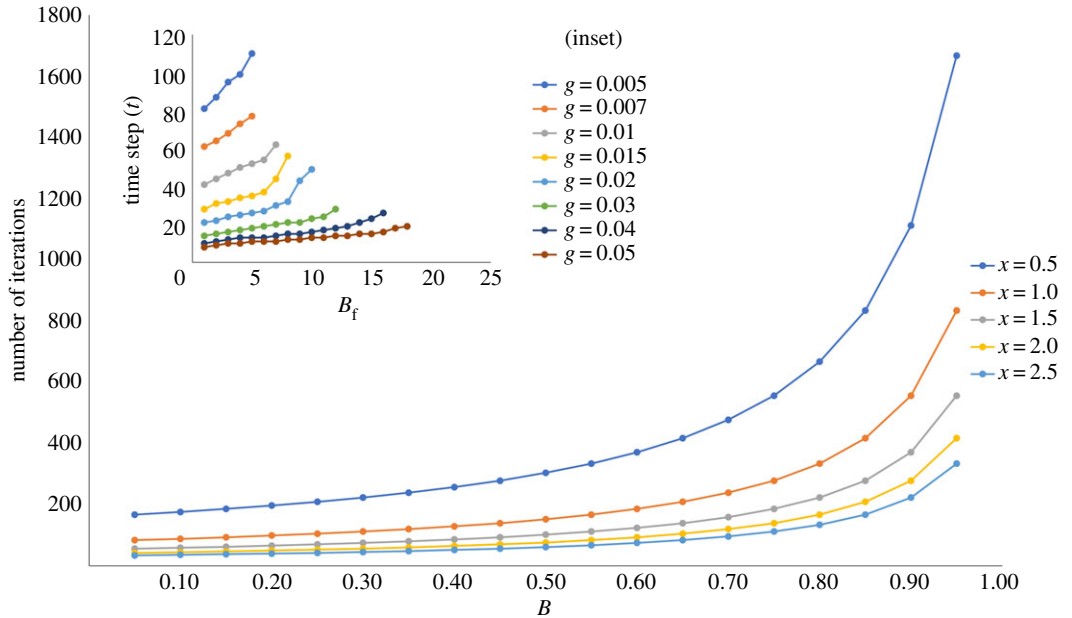

**Figure 6.** Analytical results: number of iterations versus fraction of residents breached ($B$) for $x = 0.5$, 1.0, 1.5, 2.0 and 2.5. Inset: time step versus $B_f$ outcome from computational implementation. The analytical and computational results show clear qualitative correspondence.

**Table 2.** Analytical model parameters.

| parameter | value |
|---|---|
| number of neighbourhoods ($M$) | 20 |
| number of resident agents ($N$) | 5000 |
| resident tolerance level ($\tau_r$) | 0.33 |
| migrant tolerance level ($\tau_m$) | 1.0 |

Overall, the average time to breach $\langle T \rangle$ and the fraction of breached residents $B$ over the course of the dynamics where $S$ neighbourhoods have each had $N/M$ resident tolerances breached is as follows:

$$\langle T \rangle = \frac{\sum_{s=1}^{S} T_s}{S} \tag{5.12}$$

$$\text{and} \quad B = \frac{S}{M}. \tag{5.13}$$

We now use this analytical description of the system to study system behaviour. Table 2 presents the model parameters for analytical exploration.

We plot the temporal evolution of the change in cumulative fraction of resident agents with tolerance breach. We find that the time required for the first breach to occur progressively decreases as $x$ increases, and also that the time taken for increasing fractions of residents to have their tolerances breached decreases with $x$ as evinced by the decreasing slopes of curves with increasing $x$ (figure 6). This behaviour generated by the analytical model is in close qualitative correspondence to the behaviour from our computational implementation (figure 6, inset).

Additional correspondence is evident in the inverse relationship between $B_f$ and $T_\tau$ in the computational model, and $B$ and $\langle T \rangle$ in the analytical version (figure 7a). Importantly, however, the figure also reveals a significant difference, in that the analytical model does not reproduce the nonlinearity in the relationship between $B_f$ and $T_\tau$. The nonlinearity is not replicated on account of the fact that the simple analytical model does not take into account the increasing difficulty of migrants in entering neighbourhoods with higher resident populations. The use of a probability of entry function (equation (2.2)) is critical to the dynamics of migrant entry in the simulations, where

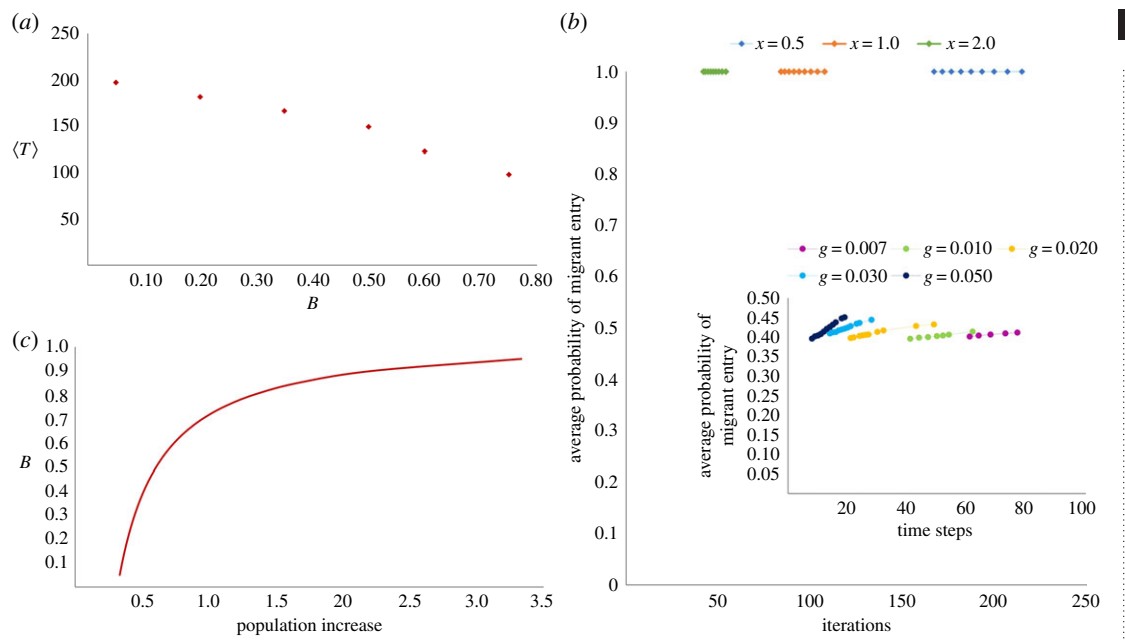

**Figure 7.** Analytical results. (a) Average time to tolerance breach ($\langle T \rangle$) versus fraction of residents with tolerance breached ($B$). There is an inverse linear relationship as opposed to the inverse nonlinear relationship in the computational interpretation. This discrepancy is due to the difference in how migrants enter the city. (b) Average probability of migrant entry versus number of iterations. These three curves represent different values of $x$ (0.5, 1, 2), and the bullets on each curve represent the number of iterations for each subsequent ($1/M$)th fraction of residents to have their tolerances breached. The probability curves are all straight lines because all migrants enter the system in the analytical model. (b) (inset): Simulation results of average probability of migrant entry over time. These five curves represent different values of $g$ (0.007, 0.01, 0.02, 0.03, 0.05), and the bullets on each curve represent time for each subsequent 1% of residents to have their tolerances breached. Probability curves get increasingly steeper with increasing $g$. (c) Fraction of residents with tolerance breached ($B$) versus population increase (as proportion of initial population). This curve is exactly the same for all values of $x$ in the analytical version, whereas in the computational model the curve varies significantly depending on value of $g$, causing the nonlinear relationship between $B_f$ and $T_\tau$.

we observe that increasing $g$ yields sharply rising average probability of migrants entering the city over time as resident tolerances are progressively breached (figure 7b (inset)), while all migrants enter and are equally distributed in the system in the analytical model (figure 7b). Indeed, when we plot overall population increase as a function of $B$ for the analytical model, we find that varying $x$ has no effect in this relationship due to the simplifying assumptions underlying the model (figure 7c), while in the computational model, varying $g$ had a significant impact on this relationship as evinced in figure 4, which we argued drove the nonlinearity observed in the relationship between $B_f$ and $T_\tau$.

# 6. Discussion

The central finding of our work therefore is that even as residents are able to move to satisfactory neighbourhoods over longer time frames (as evinced by convergence in CSI, figure 5), there are transient dynamics that potentially yield a rapid breach in tolerance of large fractions of residents in short time frames (figure 2). This phenomenon of rapid tolerance breach could potentially underlie the nature of resident responses to sharp spikes in migrants as observed in Lebanon [9] and Turkey [8], as well as in Europe [13]. The intensity of resident responses could ultimately manifest in political action in the form of mass evictions of refugees [9] or in contributing to voting patterns as described by Milanović [13].

It is important to point out that the objective of our research here is not to argue that resident attitudes towards migrants are solely shaped by rate of migrant entry. Indeed, it is apparent that resident attitudes need not necessarily be shaped by actual sharp inflows at all, but could even be due to the perception of possible inflows. For instance, in Wike et al.'s work [12] on European attitudes towards migrants, it is obvious that even countries that had very low levels of asylum applications in 2015 (the year that saw maximal refugee entry into Europe), such as Poland and Greece [7], still exhibited very negative

attitudes towards migrants. The limited point of our findings is that even in cases where refugees or migrants are accepted by nations (such as Turkey, Bangladesh, Lebanon and Germany) as a matter of humanitarian or practical principle, given the potential nonlinear increase in the proportion of residents whose tolerance levels are breached with decreasing average time to such tolerance breach, the rate of migrant entry could have a significant impact in shaping resident attitudes and potentially impact adversely the social and economic relationships between residents and migrants.

This suggests that given the short time frames available, it may well not be practically feasible for countries to accept rates of migrant entry beyond a certain level without risking social ructions. Indeed, as we have discussed earlier, this has proved to be a significant challenge for individual countries acting alone, and underlines the criticality of coordinated multi-nation responses to refugee situations. There are essential moral and ethical arguments for all major nations taking in a fair share of refugees, and this spirit animates the 1951 Refugee Convention [31], which defines the rights of refugees and the duties of states to protect them. While this essential framework is in place and has been ratified by 145 countries, it is apparent that all nations do not undertake their obligations as laid down under the convention. Our work suggests that, in addition to the moral and ethical dimension, there are practical constraints that necessitate coordinated responses. A few nations taking in large inflows, we argue, could result in significant discontent in these jurisdictions, potentially yielding negative impacts on social relationships in these societies and adversely affecting the well-being of refugees. Increased coordination in refugee intake could help prevent scenarios where migrant entry rates in individual nations (and cities) are beyond sustainable levels.

## 7. Conclusion

We model the entry of migrants into a city to study the transient dynamics generated by varying the rate of migrant entry ($g$). We find a nonlinear, inverse relationship between the fraction of resident agents whose tolerance levels have ever been breached and the average time to such tolerance breach. Beyond a certain rate of migrant entry ($g = 0.01$), we find a steep rise in the fraction of residents whose tolerance levels are breached even as the average time to breach is declining. We attempt an analytical approximation of the computational model and find qualitative correspondence in phenomenology, with significant caveats. Overall, our findings suggest the possibility that cities will realistically be unable to manage migrant influxes beyond certain velocity thresholds of entry—essentially, given the rapidity of influx, it might be difficult to manage social stability because of the high numbers of residents with tolerance breaches and the short time for this to happen. Therefore, global or coordinated multi-nation responses where refugee influx is shared between many countries and no one country (or city) takes a disproportionate share, is not just morally, but also practically, desirable.

Data accessibility. All data were simulated based on the model description in 'Model definition and specifications' section. The python code for model implementation is available at https://doi.org/10.6084/m9.figshare.7206572.
Authors' contributions. A.S. and H.J.J. conceptualized the research, chose the methodology, performed the analysis and reviewed and edited the manuscript. A.S. programmed the simulation and wrote the draft manuscript.
Competing interests. We have no competing financial interests.
Funding. We did not receive funding for this study.
Acknowledgements. A.S. gratefully acknowledges the financial support received in the form of the Schrödinger Scholarship from the Faculty of Natural Sciences, Imperial College London. The funders had no role in this research study.

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
