## [Reviewer comments · Royal Society Open Science]

Review History

RSOS-181864.R0 (Original submission)

Review form: Reviewer 1

Is the manuscript scientifically sound in its present form?

Yes

Are the interpretations and conclusions justified by the results?

Yes

Is the language acceptable?

Yes

Is it clear how to access all supporting data?

Yes

Do you have any ethical concerns with this paper?

No

Have you any concerns about statistical analyses in this paper?

No

Recommendation?

Accept as is

Comments to the Author(s)

I have studied the manuscript "Rapid migrations and dynamics of citizen response" by Sahasranaman and Jensen submitted for publication in ROOS. Here the authors explore the timely and critical problem of migrants and refugees forced to leave their homeland by using an agent-based model and also exact results obtained through a simplification of this computational model.

The authors have focused on understanding the effect of the rate of migration into cities over the residents' tolerance. Among the most important findings, they observe a rapid breach in residents' tolerance as the rates of migrant entry increases. This result thus predicts that sharp spikes in migrants may lead to rapid tolerance breach, undermining the ability of residents to receive and tolerate refugees.

The paper is well-written and the results are clearly presented. I also believe this article is a good fit for ROOS and I thus recommend its publication.

I only have two minor suggestions:

- It would be interesting to include some qualitative comments about the CSI;
- The vertical axis of Fig. 1 (right panel) could be rescaled to fit the data better.

Decision letter (RSOS-181864.R0)

23-Jan-2019

Dear Mr Sahasranaman

On behalf of the Editors, I am pleased to inform you that your Manuscript RSOS-181864 entitled "Rapid migrations and dynamics of citizen response" has been accepted for publication in Royal Society Open Science subject to minor revision in accordance with the referee suggestions. Please find the referees' comments at the end of this email.

The reviewers and handling editors have recommended publication, but also suggest some minor revisions to your manuscript. Therefore, I invite you to respond to the comments and revise your manuscript.

- **Ethics statement**

- **Data accessibility**

It is a condition of publication that all supporting data are made available either as supplementary information or preferably in a suitable permanent repository. The data

accessibility section should state where the article's supporting data can be accessed. This section should also include details, where possible of where to access other relevant research materials such as statistical tools, protocols, software etc can be accessed. If the data has been deposited in an external repository this section should list the database, accession number and link to the DOI for all data from the article that has been made publicly available. Data sets that have been deposited in an external repository and have a DOI should also be appropriately cited in the manuscript and included in the reference list.

If you wish to submit your supporting data or code to Dryad (<http://datadryad.org/>), or modify your current submission to dryad, please use the following link:
<http://datadryad.org/submit?journalID=RSOS&manu=RSOS-181864>

- **Competing interests**

- **Authors' contributions**

- **Acknowledgements**

- **Funding statement**

Because the schedule for publication is very tight, it is a condition of publication that you submit the revised version of your manuscript before 01-Feb-2019. Please note that the revision deadline will expire at 00.00am on this date. If you do not think you will be able to meet this date please let me know immediately.

To revise your manuscript, log into <https://mc.manuscriptcentral.com/rsos> and enter your Author Centre, where you will find your manuscript title listed under "Manuscripts with

Decisions". Under "Actions," click on "Create a Revision." You will be unable to make your revisions on the originally submitted version of the manuscript. Instead, revise your manuscript and upload a new version through your Author Centre.

Once again, thank you for submitting your manuscript to Royal Society Open Science and I look

forward to receiving your revision. If you have any questions at all, please do not hesitate to get in touch.

on behalf of Dr Matjaz Perc (Associate Editor) and Professor Miles Padgett (Subject Editor)
openscience@royalsociety.org

Reviewer comments to Author:
Reviewer: 1

Comments to the Author(s)

I have studied the manuscript "Rapid migrations and dynamics of citizen response" by Sahasranaman and Jensen submitted for publication in ROOS. Here the authors explore the timely and critical problem of migrants and refugees forced to leave their homeland by using an agent-based model and also exact results obtained through a simplification of this computational model.

The authors have focused on understanding the effect of the rate of migration into cities over the residents' tolerance. Among the most important findings, they observe a rapid breach in residents' tolerance as the rates of migrant entry increases. This result thus predicts that sharp spikes in migrants may lead to rapid tolerance breach, undermining the ability of residents to receive and tolerate refugees.

The paper is well-written and the results are clearly presented. I also believe this article is a good fit for ROOS and I thus recommend its publication.

I only have two minor suggestions:

- It would be interesting to include some qualitative comments about the CSI;
- The vertical axis of Fig. 1 (right panel) could be rescaled to fit the data better.

Author's Response to Decision Letter for (RSOS-181864.R0)

See Appendix A.

Decision letter (RSOS-181864.R1)

31-Jan-2019

Dear Mr Sahasranaman,

I am pleased to inform you that your manuscript entitled "Rapid migrations and dynamics of citizen response" is now accepted for publication in Royal Society Open Science.

on behalf of Dr Matjaz Perc (Associate Editor) and Prof. Miles Padgett (Subject Editor)
openscience@royalsociety.org

Follow Royal Society Publishing on Twitter: [@RSocPublishing](https://twitter.com/RSocPublishing)
Follow Royal Society Publishing on Facebook:
<https://www.facebook.com/RoyalSocietyPublishing.FanPage/>
Read Royal Society Publishing's blog: <https://blogs.royalsociety.org/publishing/>

Appendix A

Dear Dr. Perc and Prof. Padgett,

We really appreciate the referee comments on the paper and have made the changes indicated. One observation related to additional qualitative commentary on the Cell Segregation Indicator (CSI) measure, which we have now included in Section 2 (Page 5, paragraph above Eq. 2.6). The referee had also requested a rescaling of the y-axis in Fig 1 (right) and we have done this as well. We hope these changes sufficiently address the comments.

Best Regards,
Anand Sahasranaman and Henrik Jeldtoft Jensen